# The Synergistic Mechanism and Stability Evaluation of Phosphogypsum and Recycled Fine Powder-Based Multi-Source Solid Waste Geopolymer

**DOI:** 10.3390/polym15122696

**Published:** 2023-06-15

**Authors:** Xiaoming Liu, Erping Liu

**Affiliations:** School of Civil Engineering, Central South University, Changsha 410075, China; 204812349@csu.edu.cn

**Keywords:** geopolymer, multi-source solid waste, synergistic effect, volume stability, water stability, mechanical stability

## Abstract

Geopolymer prepared from solid waste is a high value-added means. However, when used alone, the geopolymer produced by phosphogypsum has the risk of expansion cracking, while the geopolymer of recycled fine powder has high strength and good density, but its volume shrinkage and deformation are large. If the two are combined, the synergistic effect of the phosphogypsum geopolymer and recycled fine powder geopolymer can realize the complementarity of advantages and disadvantages, which provides a possibility for the preparation of stable geopolymers. In this study, the volume stability, water stability and mechanical stability of geopolymers were tested, and the stability synergy mechanism between phosphogypsum, recycled fine powder and slag was analyzed by micro experiments. The results show that the synergistic effect of phosphogypsum, recycled fine powder and slag can not only control the production of ettringite (AFt) but also control the capillary stress in the hydration product, thus improving the volume stability of the geopolymer. The synergistic effect can not only improve the pore structure of the hydration product but also reduce the negative impact of calcium sulfate dihydrate (CaSO_4_∙2H_2_O), thus improving the water stability of geopolymers. The softening coefficient of P15R45 with a 45 wt.% recycled fine powder content can reach 1.06, which is 26.2% higher than P35R25 with a 25 wt.% recycled fine powder content. The synergistic work reduces the negative impact of delayed AFt and improves the mechanical stability of the geopolymer.

## 1. Introduction

Every 1 ton of phosphate fertilizer produced will produce 4.5~5 tons of industrial waste phosphogypsum, which is one of the largest discharged solid wastes in the chemical industry [1]. At present, the annual output of China’s industrial waste phosphogypsum is close to 80 million tons, the annual comprehensive utilization rate is less than 50%, and the historical stock is more than 800 million tons [2]. The main component of solid waste phosphogypsum is CaSO_4_∙2H_2_O, which is similar to that of natural gypsum and has the potential for recycling [3]. However, the utilization of phosphogypsum has a few problems. First, phosphogypsum has complex components and many impurities, which is the main factor affecting the performance of phosphogypsum products. Generally, it cannot be used directly and needs some preliminary treatment (calcination, water washing, neutralization, etc.), which not only increases the cost of phosphogypsum resource utilization but may also cause secondary pollution [4,5,6]. Second, the main chemical component of phosphogypsum is CaSO_4_∙2H_2_O. When the geopolymer is prepared with phosphogypsum, delayed ettringite (AFt) can be continuously generated, which may damage the basic structure of the geopolymer, leading to expansion damage of the material [7,8,9]. In addition, phosphogypsum contains Ba, Cr, Cu, Mn and other heavy metals, and its impact on the environment must be evaluated [10,11]. On the other hand, in China, the annual production of construction waste reaches 3 billion tons, and it is expected to reach a peak emission of 7.5 billion tons by 2035 [12,13]. The open-air stacking of massive construction waste occupies a large amount of urban spatial resources, causing serious pollution to the air and groundwater. Construction waste is divided into three categories through crushing and screening: recycled coarse aggregate (over 4.75 mm), recycled fine aggregate (0.15–4.75 mm), and recycled fine powder (below 0.15 mm). In recent years, a large number of studies have shown the applicability of construction waste and recycling coarse and fine aggregates in the production of recycled concrete, recycled mortar, floor tiles and road base [14,15,16,17,18], and some of the results have been applied to engineering practice. However, due to the characteristics of high porosity, high water absorption and low density, research is less and the utilization of recycled fine powder is difficult. The chemical composition of construction waste recycled fine powder is very close to that of fly ash, which contains a large amount of SiO_2_ and Al_2_O_3_ and has the potential as an auxiliary cementitious material [19]. Zhang et al. [20] studied the properties of recycled fine powder via single or compound addition and found that the mechanical properties of cement mortar with a single addition of 10 wt.% recycled fine powder were the best. When the amount of recycled fine powder exceeds 10 wt.%, a part of fly ash can be added to improve the mechanical properties of cement mortar. Xiao et al. [21] found that once the amount of recycled fine powder exceeds 30 wt.%, it can cause severe shrinkage, thereby affecting the mechanical properties of concrete. Yang [22] found that there is a synergistic effect between gypsum slag cement and recycled concrete fine powder, which can significantly improve the unfavorable development of recycled concrete with fine powder when used together.

Geopolymers have many advantages, such as fast hardness and high strength, acid resistance, alkali resistance, low carbon and environmental protection. It is considered the most potential low-carbon green cementitious material to replace cement; therefore, geopolymer has received the attention of many scholars at home and abroad as soon as it appeared. In the 1960s, Davidovits, a French scholar, summarized the experiences of his predecessors and proposed the concept of a geopolymer for the first time [23]. Although there is still disagreement in academic circles regarding the reaction mechanism of geopolymers, most scholars agree that the basic reaction mechanism of geopolymers is that the silicon–oxygen and aluminum–oxygen bonds in geopolymer precursor materials break under an alkaline environment to form AlO_4_ and SiO_4_ tetrahedral monomers, and then undergo polymerization and reorganization to form a three-dimensional mesh-like gel [23,24,25,26]. The preparation of geopolymers using bulk solid wastes with silica-aluminous as the main component, such as metakaolin, slag, fly ash and rice husk ash, is a current research hotspot [24,27,28,29]. The shrinkage problem of geopolymers is one of the main factors affecting their practical engineering applications [30,31]. Severe shrinkage can cause overall shrinkage stress and deformation of the material. If the material deforms unevenly, it will cause harmful cracks, further reducing the service life of the material and affecting structural safety. The main sources of geopolymer shrinkage are as follows. First, the mesopores in geopolymers account for a large proportion of the slurry pores. In a dry environment, the loss of water in the pores generates significant capillary stress [32], resulting in significant shrinkage. Second, the hydration products in the geopolymer gel are mainly amorphous NASH, CSH and CASH, and these hydration products shrink when the humidity decreases to a certain extent [33]. Third, the geopolymer contains residual unreacted sodium silicate that reacts with CO_2_ to generate silica gel with high water content, which also dehydrates and shrinks under dry conditions [34]. At present, the shrinkage performance of alkali-activated cementitious materials is mainly improved by the types of activators, mineral admixtures, chemical additives, fiber-reinforced materials, porous aggregates and curing conditions [35]. Research has shown that recycled aggregate can be used as an “internal curing agent” to improve the drying shrinkage performance of alkali-activated slag based on its high water absorption [36]. On the other hand, research has shown that the strength of geopolymers can be improved by incorporating precursor materials with high calcium content and high reactivity, which will be beneficial for improving the matrix’s ability to resist dry shrinkage deformation [37]. However, many studies [38,39] indicated that the addition of precursor materials with high calcium content increased the proportion of mesoporous pores in the slurry, thereby increasing the dry shrinkage of the specimen. It is believed that the pore size distribution and deformation modulus of composite materials jointly determines the drying shrinkage rate of the specimen. Only when the strength increase is relatively slow, the influence of the pore structure on drying shrinkage is more significant [40]. If the recycled fine powder is combined with solid wastes, such as phosphogypsum and slag, to prepare the geopolymer, on the one hand, slag can improve the strength of the slurry, reduce the drying shrinkage of recycled fine powder geopolymer and enable the geopolymer to obtain early strength at room temperature. On the other hand, recycled phosphogypsum and fine powder can be used as “internal curing agents” to reduce the drying shrinkage of slag components in the slurry [36].

In this study, many types of original solid wastes, such as original phosphogypsum and recycled fine powder, were used as the main raw materials; slag was used as the active blending material; modified sodium silicate solution was used as the alkali activator; and the stable synergistic effect between the original phosphogypsum, recycled fine powder, slag and other solid wastes were used to prepare the geopolymer. The hydration mechanism and the stable synergistic mechanism between the solid wastes were analyzed. First of all, the main component of raw phosphogypsum is CaSO_4_∙2H_2_O, which can participate in the hydration reaction to form the expansion phase AFt crystal in an alkaline environment, which can not only improve the pore structure of the geopolymer but also offset part of the volume shrinkage deformation of the geopolymer. Secondly, the recycled fine powder can not only promote the formation of hydrated gels, such as calcium aluminosilicate hydrate (CASH) and sodium aluminosilicate hydrate (NASH), but also has the skeleton effect and pozzolanic effect, which can refine the pore structure of the geopolymer, leading to an increase in capillary stress, thus increasing the volume shrinkage deformation of the geopolymer, and offsetting the volume expansion deformation caused by AFt. At the same time, soluble *P*, *F* and other impurities in phosphogypsum are converted into inert substances, reducing their negative impact on cementitious materials. The geopolymer has a positive significance in promoting the resource utilization of phosphogypsum and recycled fine powder.

## 2. Raw Materials and Experimental Methods

### 2.1. Raw Materials

Phosphogypsum (PG): Phosphogypsum, taken from a phosphate fertilizer factory in Guiyang, China, is mainly composed of CaSO_4_∙2H_2_O, with a pH value of about 4.8 (solid water ratio of 1:10), the water of crystallization content of about 16.6% and a natural water content of more than 13.1%, which is dark gray. Its appearance is shown in Figure 1a. Phosphogypsum was dried at 40 °C to a constant weight, crushed manually, and passed through a 0.15 mm square mesh sieve to obtain phosphogypsum powder with particle sizes of 0.15 mm or less. See Figure 1b for details. The main chemical composition, phase analysis results and the microstructure of PG are shown in Table 1, Figure 3 and Figure 4a, respectively. As shown in Table 1 and Figure 3, the main component in PG is CaSO_4_∙2H_2_O, containing a small number of impurities such as quartz.

Recycled construction waste fine powder (RP): The recycled brick and concrete fine aggregate used in the experiment was obtained from Yunzhong Construction Waste Recycling Company in Changsha, China, with a particle size range of 0–4.75 mm. Among them, the proportion of waste clay bricks and mortars exceeded 80%. The laboratory preparation method of RP is shown in Figure 2. First, the recycled brick mixed with fine aggregate was dried to a constant weight at 105 °C and passed through a 2.36 mm sieve. Then, it was crushed separately using a small laboratory crusher. Finally, it was mixed evenly and passed through a 0.15 mm sieve to obtain a fine powder with a particle size of less than 0.16 mm, known as RP. The main chemical composition, phase analysis results and microstructure of RP are shown in Table 1, Figure 3 and Figure 4b, respectively. From Table 1 and Figure 2, it can be seen that the main chemical components of RP are Al_2_O_3_ and SiO_2_, with a combined proportion of over 65%.

Slag: S95 grade blast furnace slag produced by the Hengyuan Material Factory in Zhengzhou, China, has a specific surface area of 412 m^2^/kg, a 28-day activity index of 95.5% and an alkalinity coefficient of 1.058. The main chemical components are shown in Table 1.

Ordinary Portland cement (OPC): The cement used in the experiment is P.O42.5 ordinary Portland cement provided by Shuoshun Cement Company in Weifang City, Shandong, China. The technical indicators are shown in Table 2.

Sand (S): The sand used in the experiment is fine river sand, and its particle size distribution is shown in Table 3.

Sodium hydroxide: The sodium hydroxide used in the experiment is a sheet-like NaOH provided by Zhongtai Chemical Company in Xinjiang, China, with content greater than 99.0%. It is mainly used to adjust the modulus of the modified sodium silicate solution.

Sodium silicate solution: It is a liquid sodium silicate solution with a modulus of 3.2 kg/cm^3^ and water content of 65% was used.The related parameters can be found in Table 4.

Water-reducing agent: The water-reducing agent used in this experiment is yellow powdered naphthalene.

Water (W): The experimental mixing water is laboratory tap water.

### 2.2. Mix Proportion Design

In order to investigate the effects of various powder material contents on the volume change, porosity and pore saturation of the geopolymer, 8 groups of experiments with different powder material contents were set up, with the modulus of the alkali activator and alkali equivalent fixed at 1.3 and 6%, respectively. In the test group with slag content of 40 wt.%, five kinds of phosphogypsum content were set, namely 0 wt.%, 15 wt.%, 25 wt.%, 35 wt.% and 60 wt.%. In the test group with the 25 wt.% phosphogypsum content, four kinds of slag content were set, namely, 20 wt.%, 30 wt.%, 40 wt.% and 50 wt.%. At the same time, a group of pure cement slurry specimens was set up as the control group for comparative analysis. The content in the test was calculated according to the total mass of the powder materials (phosphogypsum, recycled fine powder and slag), and the water–binder ratio was fixed at 0.42. The specimens were numbered according to the content of each powder material and the parameters of the alkali activator. For example, for the specimen numbered P25R35 (R35S40), P25 indicated that the content of phosphogypsum was 25 wt.%, R35 indicated that the content of recycled fine powder was 35 wt.%, and S40 indicated that the content of slag was 40 wt.%. The control group was directly named OPC, and the specific experimental mix ratio is shown in Table 5.

In order to explore the influence of recycled fine powder content and slag content on the water stability and mechanical stability of the geopolymer, five groups of mix proportions with different powder material contents were used; the slag content was fixed at 40 wt.% and the recycled fine powder content was 25 wt.%, 35 wt.% and 45 wt.%, respectively. When the recycled fine powder content was fixed at 35 wt.%, the slag content was 30 wt.%, 40 wt.% and 50 wt.%, respectively. In the mortar test, the content of each powder was also calculated based on the total mass of the powder material, with the modulus and alkali equivalent fixed at 1.3 and 6%, water–cement ratio fixed at 0.48 and cement–sand ratio fixed at 1:2. At the same time, in order to facilitate the formation of mortar specimens and meet practical application needs, all mortar specimens were mixed with 1 wt.% powder water-reducing agent (calculated based on the total mass of powder) to improve the fluidity of their mortar. The numbering rules for the mortar specimens are the same as those in Table 5, and the specific mix design is detailed in Table 6. It should be noted that in the mechanical stability experiment, the standard cured 90-day mortar specimens and the soaked cured 90-day mortar specimens were marked with (S90) and (W90), respectively, after their respective numbers to distinguish them.

### 2.3. Specimen Preparations and Curing Conditions

(1)Forming and curing of geopolymer slurry specimens: First, accurately weighed various powder materials were taken according to the mix proportion in the experiment, mixed and stirred evenly, and then poured together with the prepared modified sodium silicate solution into the mortar mixer. Second, it was first slowly stirred for 1 min, after which the bottom slurry of the pot was scraped manually stirred for 1 min, and then stirred continuously for 2 min to ensure that the powder material and solution were evenly mixed. Then, the inner wall of the mold (25 mm × 25 mm × 280 mm) was cleaned and brushed with engine oil, and then the stirred slurry was poured into the mold and vibrated 60 times, the mold flat was scraped, and the specimens were wrapped and sealed with a cling film. The samples were cured at room temperature (20 ± 3 °C) for 24 h. Finally, the specimen was completely removed from the mold and marked. Then, the specimen was sealed and wrapped with a cling film and transferred to a cement curing box with constant temperature curing until the specified age. It should be noted that some specimens with longer setting times needed to be cured for 48 h before demolding.(2)Forming and curing of geopolymer mortar specimens: The formation and curing process of the geopolymer mortar specimens was the same as that of the pure slurry specimens. The mortar forming size was 40 mm × 40 mm × 160 mm.

### 2.4. Tests

(1)Volume change rate test: This test was conducted in accordance with the “Test Method for Dry Shrinkage of Cement Mortar” (JC/T 603-2004) [41]. After 24 h of sealing and curing, the initial length of the specimen was measured using a BC-300 cement length meter (see Figure 5) and then placed in an environment of 20 ± 3 °C and 50 ± 5% relative humidity for 154 days. Measurements were taken every 24 h in the first two weeks; every 48 h from the 3rd to the 4th week; every 7 days from the 5th to the 8th week; every 14 days from the 9th to the 22nd week. The average of the results from each group of 3 specimens was used for the analysis. The volume change rate of the specimen was calculated using Equation (1) as follows:(1)$$S_{n}=\frac{\left(L_{n}-L_{0}\right)\times 100}{250}$$
where $S_{n}$: rate of volume change, contraction is positive and expansion is negative; %; $L_{0}$: initial length of the specimen, mm; $L_{n}$: the measured length of the specimen cured at 20 ± 3 °C and 50 ± 5% relative humidity for *n* days, mm.(2)Porosity and porosity saturation tests: First, the mass of the slurry specimen was weighed and its volume was measured, which were recorded as the mass and volume of the specimen, respectively. The slurry specimen was broken into small pieces and immersed in clean water for 24 h to ensure that the pieces were saturated with water. The surface of the fragments was removed and dried, and the total mass of the fragments was weighed and recorded as the saturated mass of the specimen. Finally, the fragments were placed in an oven (50 °C) for drying, and the total mass of the fragments was weighed at this time and recorded as the dry mass of the test piece. The porosity of the specimen was calculated using Equation (2), and the pore saturation of the specimen was calculated using Equation (3) as follows:(2)$$P=\frac{M_{1}-M_{2}}{\rho_{w}V}\times 100\%$$
(3)$$S=\frac{M-M_{2}}{M_{1}-M_{2}}\times 100\%$$
where P: the porosity of the specimen, %; S: the saturation of pores, %; M: mass, g; M_1_: saturation mass, g; M_2_: dry mass, g; V: volume, cm^3^; $\rho_{w}$: density of water, 1 g/cm^3^.(3)Compressive strength test: The standard “ISO method for testing the strength of cement mortar” (GB/T 17671-1999) was referred to for the testing [42].(4)Softening coefficient test: Reference [43] was referred to for the testing. Two sets of mortar specimens with a curing age of 28 days were taken, and one set was dried in an oven at (45 ± 2) °C to a constant weight. Its compressive strength was measured and recorded as dry the compressive strength. The other group was immersed in warm water at (20 ± 3) °C for 24 h, and then the surface was taken out and wiped dry, and its compressive strength was measured and recorded as the wet compressive strength. The formula for calculating the softening coefficient is shown in Equation (4).
(4)$$S_{\delta }=\frac{\sigma_{g}}{\sigma_{s}}$$
where $S_{\delta }$: softening coefficient; $\sigma_{g}$: dry compressive strength, Mpa; $\sigma_{s}$: wet compressive strength, Mpa. (5)Mechanical stability performance test: The measurement method of this test was conducted according to reference [44]. Two sets of specimens were taken from the mortar specimens with a curing age of 28 days. One group of specimens was subjected to standard curing at a temperature of (20 ± 3) °C and a relative humidity of (90 ± 5)%, while the other group was subjected to immersion curing in water at a constant temperature of 20 ± 3 °C. After 90 days, the compressive strength of the two sets of specimens was tested separately.(6)X-ray fluorescence spectrometer (XRF): Powder samples (3 g) were prepared and an X-ray fluorescence spectrometer with Panalytical Axios model was used, and the test mode was in the form of an oxide.(7)X-ray diffraction (XRD): The specimen at a specified age was first crushed into small pieces. The pieces <1.18 mm were taken and immersed into anhydrous alcohol for 7 days to terminate the hydration process, and then dried at 50 °C to a constant weight. Finally, the samples were ground into a powder and 1 g was selected for the XRD test after being passed through a 200 mesh sieve. Bruke D8 advanced X-ray diffractometer with a scanning speed of 2°/min and scanning angle range of 5~90° was adopted.(8)Fourier-transform infrared spectroscopy (FTIR): The sample preparation method was the same as that of the XRD. The test mode was a conventional powder tablet. The wavenumber range was 400~4000 cm^−1^.(9)Scanning electron microscope (SEM): The block sample with a thickness of less than 1 cm, diameter ≤ 1 cm and relatively flat fracture surface was selected, dried at 50 °C to constant weight, and the Czech Tescan Mira LMS scanning electron microscope was used to test the morphology.

## 3. Discussion

### 3.1. Volume Stability Analysis

Volume stability is one of the main factors affecting the practical application of cementitious materials, especially for alkali-activated materials, and volume change is an effective indicator to measure volume stability. At present, widely accepted theories include the interlayer water theory, solid surface energy theory, capillary pressure theory, etc. The capillary pressure theory is applicable to the volume change of alkali-activated cementitious material paste, so the capillary pressure theory is also applicable to the phosphogypsum and recycled fine powder-based multi-source solid waste geopolymer.

According to capillary theory, the key indicators that affect the volume change of conventional recycled fine powder-based geopolymers include the bulk modulus *K*, pore size *r* and pore saturation *S* [45,46,47,48]. At the same time, phosphogypsum participates in the hydration reaction in an alkaline environment to generate the expansive phase ettringite (AFt), whose solid volume expands by about 1.25 times, which can effectively offset some volume shrinkage changes of the slurry [9]. Therefore, the key indicators affecting the volume stability of geopolymers mainly include the bulk modulus *K*, pore diameter *r*, pore saturation *S* and the quantity of AFt. The bulk modulus *K* can be calculated using Equation (5), and the elastic modulus *E* is related to the compressive strength *σ.* The relationship between *E* and *σ* can be calculated using Equation (6) [49]. In addition, there is a significant positive correlation between pore *r* and porosity *P.* Therefore, the key index affecting the volume stability of geopolymers can be expressed as the compressive strength *σ*, porosity *P*, porosity saturation *S* and the quantity of AFt.
(5)$$K=\frac{E}{3\left(1-2\upsilon \right)}$$
(6)$$E=5700\sqrt{\sigma }$$

#### 3.1.1. Effect of Phosphogypsum on Volume Stability

(1)Volume change rate

Figure 6 shows the relationship between the 154-day volume change rate of phosphogypsum and recycled fine powder-based multi-source solid waste geopolymer and the phosphogypsum content. It can be seen from Figure 6 that the volume change rate of the geopolymer decreases with an increase in the phosphogypsum content. The 154-day volume change rates of specimens OPC, P0R60, P15R45, P25R35, P35R25 and P60R0 are 0.21%, 0.88%, 0.63%, 0.43%, 0.30% and 0.13%, respectively. The 154-day volume change rates of specimens P15R45, P25R35 and P35R25 are reduced by 27.8%, 51.8% and 66.4%, respectively, compared to the 154-day volume change rates of specimens P0R60. It shows that phosphogypsum can improve the volume stability of geopolymers. On the one hand, the effect of phosphogypsum content on the porosity *P* and pore saturation *S* of the slurry is relatively small (see the analysis in Figure 7), so the volume deformation caused by capillary stress changes in the slurry due to the change in phosphogypsum content is small. On the other hand, with the increase in phosphogypsum content, the production of AFt increases significantly, which counteracts the bulk shrinkage deformation of the slurry. Therefore, the volume change rate of the slurry decreases with an increase in phosphogypsum content. From Figure 6, it is not difficult to see that the 154-day volume change rate of specimen P35R25 is only 39.5% higher than that of specimen OPC. Meanwhile, the 28-day volume change rate of specimen P35R25 is 10.7% lower than that of specimen OPC. This shows that phosphogypsum and recycled fine powder-based multi-source solid waste geopolymers have good volume stability.

(2)Porosity and pore saturation

Figure 7 shows the relationship between the porosity and pore saturation of phosphogypsum and recycled fine powder-based multi-source solid waste geopolymers and the content of phosphogypsum. As shown in Figure 7a, the porosity of specimens P0R60, P15R45, P25R35, P35R25 and P60R0 are 8.5%, 10.3%, 9.7%, 11% and 11.2%, respectively. Compared to specimen P0R60, the porosity of P15R45, P25R35, P35R25 and P60R0 increase gradually with an increasing phosphogypsum content, indicating that phosphogypsum will destroy the pore structure of the geopolymer, leading to an increase in the porosity of the slurry. For all specimens mixed with phosphogypsum, the porosity first decreases and then increases with an increase in phosphogypsum content. This shows that properly increasing the content of phosphogypsum will not destroy the pore structure of the slurry, but it will improve the density of the slurry, leading to a reduction in the porosity of the slurry. It can be observed from Figure 7b that the pore saturation of the slurry first increases and then decreases with the phosphogypsum content, which is completely opposite to the porosity of the slurry. At the same time, it can be seen in Figure 7 that the maximum difference in slurry porosity is only 1.7%, and the maximum difference in slurry pore saturation is only 13%, indicating that the content of phosphogypsum has little influence on the porosity and pore saturation of the geopolymer.

#### 3.1.2. Effect of Slag on Volume Stability

(1)Volume change rate

Figure 8 shows the relationship between the 154-day volume change rate of phosphogypsum and recycled fine powder-based multi-source solid waste geopolymer and the slag content. As shown in Figure 8, the volume change rate of the geopolymer at 154 days first decreases and then increases with an increase in the slag content. The volume change rates of specimens R55S20, R45S30, R35S40 and R25S50 at 154 days are 0.62%, 0.49%, 0.43% and 0.82%, respectively. The volume change rates of specimens R45S30 and R35S40 at 154 days decrease by 21.2% and 31.0%, respectively, compared to specimens R55S20. This indicates that slag can improve the volume stability of geopolymers. It can be assumed that the slag promotes the hydration reaction in the system, generating more hydrated hydration, such as CSH, CASH, NASH and Aft, which makes the slurry denser and improves its compressive strength, thus enhancing the ability of the slurry to resist volume deformation [50]. In addition, more Aft can also offset more slurry volume shrinkage deformation. On the other hand, slag can refine the pore structure of the slurry, reducing its porosity *P* while increasing its pore saturation *S*, resulting in an increase in capillary stress within the system (see Figure 9) [49]. Obviously, the influence of the bulk modulus *K* and the amount of AFt generated on the volume deformation of the slurry dominates, so the volume change rate of the specimen decreases with an increase in the slag content. Meanwhile, Figure 8 shows that the 154-day volume change rate of specimen R25S50 is 93.2% higher than the 154-day volume change rate of specimen R35S40. This shows that it cannot improve the volume stability of the geopolymer when the slag content is too high, but it will damage its volume stability. This is because excessive slag improves the porosity *P* of the slurry and reduces the pore saturation *S* of the slurry, resulting in a reduction in the capillary stress. Excessive slag also inhibits the hydration reaction, resulting in a decrease in the compressive strength *σ* and the production of AFt. However, the production of compressive strength *σ* and AFt is still dominant, and the volume change rate of the slurry increases.

(2)Porosity and pore saturation

Figure 9 shows the relationship between the porosity and pore saturation of phosphogypsum and recycled fine powder-based multi-source solid waste geopolymer and slag content. As shown in Figure 9a, the porosity of specimens R55S20, R45S30, R35S40 and R25S50 are 16.7%, 13.0%, 9.7% and 10.8%, respectively. The porosity of specimens R45S30 and R35S40 decrease by 22.2% and 41.9%, respectively, compared to that of specimen R55S20. This indicates that the slag can improve the density of geopolymers and reduce the porosity of the slurry. However, the porosity of specimen R25S50 increased by 11.3% compared to that of specimen R35S40, indicating that an appropriate amount of slag is required; otherwise, it will damage the pore structure of the geopolymer and lead to an increase in the porosity of the slurry. From Figure 9b, it can be observed that the pore saturation of specimens R55S20, R45S30, R35S40 and R25S50 are 31%, 60%, 77% and 74%, respectively. The pore saturation of specimens R45S30 and R35S40 increased by 193.5% and 148.4%, respectively, compared to that of specimen R55S20, whereas the pore saturation of specimen R25S50 decreased by 3.9% compared to that of specimen R35S40. This indicates that the pore saturation of the polymer in the area first increases and then decreases with an increase in the slag content. Meanwhile, as shown in Figure 9, the maximum difference in slurry porosity is 7.0%, and the maximum difference in slurry pore saturation is 46%, indicating that the slag content significantly affects the porosity and pore saturation of geopolymers.

### 3.2. Water Stability Analysis

Water stability is a performance index that represents the water resistance of cementitious materials and determines whether the cementitious materials can be safely used in water or wet environments. However, the water stability of raw phosphogypsum-based cementitious materials is generally poor, so the water stability of phosphogypsum and recycled fine powder-based multi-source solid waste geopolymers must be evaluated. The softening coefficient is the main effective index to measure water stability.

Figure 10 shows the relationship between the water stability of phosphogypsum and recycled fine powder-based multi-source solid waste geopolymer and the amount of recycled fine powder. As shown in Figure 10, the softening coefficients of mortar specimens P35R25, P25R35 and P15R45 increase successively by 0.84, 0.98 and 1.06, respectively. This indicates that the water stability of the geopolymer increases with an increase in the recycled fine powder. 

#### 3.2.1. Effect of Recycled Fine Powder on Water Stability

It can be seen from the analysis that a large number of phosphogypsum particles with incomplete reaction must remain in mortar specimen P35R25 (which can be proven by the XRD diagram), whereas the solubility of the main components ${\mathrm{CaSO}}_{4}\cdot 2H_{2}O$ in phosphogypsum is large, which easily dissolves in water and recrystallizes, leading to a decrease in the strength of the matrix [51]. Therefore, the wet compressive strength of mortar specimen P35R25S40M1.3N is significantly lower than its dry compressive strength. The softening coefficient of mortar specimens increases with an increase in the recycled fine powder content. This is because the increase in the amount of recycled fine powder promotes the formation of more CASH, NASH and other hydrated gels in the matrix. These products will not only cover the surface of phosphogypsum particles and prevent them from contacting water but also improve the density of the matrix, reduce the porosity and block the entry of external water, thereby reducing the negative impact. On the other hand, with a decrease in the phosphogypsum content, the residual phosphogypsum particles in the matrix will also be reduced, resulting in a negative impact of dissolution and recrystallization on the matrix, which will also be significantly reduced. The final performance is that the water stability of the geopolymer significantly improves with an increase in the recycled fine powder content.

The softening coefficients of mortar specimens P15R45 and P25R35 are both greater than 0.85, whereas the softening coefficient of mortar specimen P35R25 is only 0.01 less than 0.85, indicating that the geopolymer has good water stability. It is worth noting that the soft coefficient of mortar specimen P15R45 is greater than 1, indicating that its wet compressive strength is greater than its dry compressive strength. This is because after 28 days of curing with the sealing film, the hydration reaction inside the matrix still does not stop, and both the constant temperature (40 °C) drying and immersion wetting test methods are unable to prevent the hydration reaction inside the system from continuing. Obviously, immersion curing can promote the hydration reaction of geopolymers and generate more hydration products, thereby promoting strength growth [52]. In addition, it should be noted that the naphthalene water reducer can not only accelerate the rate of the hydration reaction and reduce the generation of delayed AFt but also change the morphology of AFt crystals from elongated to short thickness, which can improve the compactness of the matrix. Therefore, naphthalene water reducers have important effects on the mechanical properties and water stability of the polymers.

#### 3.2.2. Effect of Slag on Water Stability

Figure 11 shows the relationship between the water stability of phosphogypsum and recycled fine powder-based multi-source solid waste geopolymer and the slag content. As shown in Figure 11, the softening coefficients of mortar specimens R45S30, R35S40 and R25S50 increase sequentially, with values of 0.92, 0.98 and 0.99, respectively. The water stability of the geopolymer increases with an increase in the slag content. This is because slag promotes hydration reactions and generates more hydration products. This improves the compactness of the matrix, reduces the porosity and reduces the possibility of water entering the matrix from the pores. On the other hand, more phosphogypsum particles (less) are consumed, which reduces the residual phosphogypsum particles in the matrix and reduces the negative impact on the matrix. Therefore, slag can reduce the harm caused by dissolution and recrystallization to the matrix, thereby improving water stability. The softening coefficient of mortar specimen R25S50 is 8.2% higher than that of mortar specimen R45S30. The softening coefficient of mortar specimen P35R25 decreases by 20.1% compared to that of mortar specimen P15R45. This shows that the ratio of recycled fine powder to phosphogypsum has a greater influence on the water stability of geopolymers than the ratio of slag to recycled fine powder.

### 3.3. Mechanical Stability Analysis

Mechanical stability is an effective indicator to measure the stability of the strength of cementitious materials over a long period of time, and it is a key factor for the long-term and safe application of cementitious materials in various engineering practices. Based on the characteristics of phosphogypsum and recycled fine powder-based multi-source solid waste geopolymers, the standard curing compressive strength and hydrating compressive strength of a longer age were used to measure its mechanical stability.

#### 3.3.1. Effect of Recycled Fine Powder on Mechanical Stability

Figure 12 shows the relationship between the mechanical stability of phosphogypsum and recycled fine powder-based multi-source solid waste geopolymers and the amount of recycled fine powder. From Figure 12, it can be seen that after 28 days of sealing and curing under standard conditions for 90 days, the compressive strength of mortar specimens P35R25 (S90), P25R35 (S90) and P15R45 (S90) are 11.99, 29.26 and 32.48 MPa, respectively, indicating a decrease of 23.3%, 13.8% and 2.3% compared to 90 days ago. This indicates that there is a risk of instability and a sudden decrease in the strength of geopolymers in the later stage, and recycled fine powders can effectively alleviate this trend of mechanical properties. This is because the hydration reaction inside the matrix will not completely stop after 28 days of sealing and curing, and it will continue to react to generate hydration products in the subsequent aging period. Within 90 days of standard curing, the internal reaction of the system continues to generate delayed AFt and some products, such as CSH, CASH and NASH. CSH, CASH, NASH and other gels are beneficial to the structure and strength of the matrix, whereas delayed AFt may damage the structure of the matrix, resulting in a significant decrease in the strength of the matrix. Obviously, for phosphogypsum and recycled fine powder-based multi-source solid waste geopolymers, the influence of delayed AFt on the later structure and strength of the geopolymer is far greater than that of CSH, CASH, NASH and other hydration gels, and occupies a dominant position. Therefore, the compressive strength of geopolymers will decrease due to the destructive effect of the delayed AFt on the structure. Phosphogypsum is the main internal source of recycled fine powder-based multi-source solid waste geopolymers, and it is the key to controlling the amount of AFt generated in the system [8]. Therefore, the negative effect of delayed AFt on the geopolymer decreases with a decrease in the phosphogypsum content. The increase in recycled fine powder content leads to a decrease in the phosphogypsum content, which not only strengthens the positive effect of CSH, CASH, NASH and other hydration products on the structure but also reduces the negative effect of delayed AFt, which is ultimately manifested in the reduction in standard curing compressive strength. As shown in Figure 13a, mortar specimen P35R25 (S90) after 90 days of standard curing has obvious cracks on the surface, which is because the amount of recycled fine powder is small, resulting in less hydration and gelling products. In addition, the amount of phosphogypsum is large, resulting in a strong negative effect of the delayed AFt, which ultimately leads to the destruction of the matrix structure and cracking. In conclusion, recycled fine powder can improve the standard curing strength of geopolymers for a long time.

From Figure 12, it can be seen that after 28 days of sealing and curing, and then soaking in water at room temperature for 90 days, the compressive strength of mortar specimens P35R25 (W90), P25R35 (W90) and P15R45 (W90) are 14.77, 30.47 and 38.41 MPa, respectively, which are 23.2%, 4.1% and 18.2% higher than the compressive strength of mortar specimens after 90 days of standard curing. The results show that the hydroponic conditions are more conducive to the strength development of geopolymers. This is because there are some pores in the mortar specimen soaked in water (as shown in Figure 9), and external water can slowly penetrate the interior of the mortar specimen through the pores. On the one hand, the water that infiltrates into the system can promote hydration reactions and generate more hydration products, such as CSH, CASH and NASH, which have a positive effect on the structure and strength of the matrix. On the other hand, it has a high solubility in water, which can be lost from the matrix with water, reducing the concentration inside the system, and thus reducing the negative effect of the delayed AFt on the matrix; as a result, the structure and strength of the matrix will improve. In addition, as shown in Figure 12, after 90 days of soaking in water at room temperature, the compressive strength of mortar specimen P25R35 decreased by 10.2% compared to the compressive strength before 90 days, whereas the compressive strength of mortar specimen P15R45 increased by 15.5% compared to the compressive strength before 90 days. This indicates that recycled fine powder can also improve the long-term water-curing strength of geopolymers. The main reason is that the increase in recycled fine powder will lead to an increase in CSH, CASH, NASH and other gelling products, whereas a decrease in phosphogypsum content will lead to a decrease in delayed AFt production. It should be noted that after 90 days of water curing, the compressive strength of mortar specimen P35R25 (W90) decreases by only 5.5% compared to the compressive strength before 90 days. However, this does not mean that the long-term water stability of mortar specimen P35R25 (W90) is stronger than that of mortar specimen P25R35 (W90). This is because mortar specimen P35R25 contains less recycled fine powder and more phosphogypsum. After 28 days of sealed curing, its structure has been damaged. In summary, recycled fine powders can enhance the mechanical stability of geopolymers. From Figure 13, it can be observed that there are no obvious cracks in mortar specimens P15R45 (S90), P15R45 (W90), P25R35 (S90) and P25R35 (W90), and the appearance is very complete. The structures of mortar specimens P35R25 (S90) and P35R25 (W90) are both damaged, with obvious cracks appearing, and the crack gap of mortar specimen P35R25 (W90) is larger. Therefore, the mechanical stability of mortar specimen P35R25 at a longer age is relatively poor, which is consistent with the analysis in Figure 12.

#### 3.3.2. Effect of the Slag on the Mechanical Stability

Figure 14 shows the relationship between the mechanical stability of phosphogypsum and recycled fine powder-based multi-source solid waste geopolymers and the slag content. From Figure 14, it can be seen that after 28 days of sealing and curing under standard conditions for 90 days, the compressive strength of mortar specimens R45S30 (S90), R35S40 (S90) and R25S50 (S90) are 11.12, 29.26 and 16.76 MPa, respectively, indicating a decrease of 36.9%, 13.8% and 52.5% compared to 90 days ago. Adding slag can improve the standard curing strength of geopolymers over a longer period of time. On the one hand, during the generation process of AFt, expansion stress will be generated due to its expansion; on the other hand, the matrix will generate binding forces due to strength development [53]. For mortar specimen R45S30, the binding force generated by its strength development is insufficient to offset the expansion stress generated by AFt, resulting in the destruction of the matrix structure (see Figure 15a) and a significant decrease in strength. With an increase in the slag content, more hydration products, such as CSH, CASH and NASH, are promoted, significantly enhancing the strength, leading to a significant increase in the binding force and offsetting the majority of the expansion stress, thus enhancing the long-term mechanical properties of the matrix. However, when the amount of slag is too large, its strength constraint continues to increase while also refining the pore structure of the matrix. At this point, the pores in the matrix are small, leading to a sharp increase in the expansion stress generated by AFt, leading to the failure of the matrix structure (see Figure 15) and a significant decrease in strength. 

As shown in Figure 14, after 28 days of membrane curing and 90 days of immersion curing, the compressive strength of mortar specimens R45S30 (W90), R35S40 (W90) and R25S50 (W90) are 22.18, 30.47 and 18.27 MPa, respectively. The immersion compressive strength of mortar specimens R45S30 (W90) increased by 25.9% compared to the compressive strength before 90 days, whereas the immersion compressive strength of mortar specimens R35S40 (W90) and R25S50 (W90) decreased by 10.2% and 48.2%, respectively, compared to their compressive strength before 90 days. The water-curing strength of the geopolymer decreased with an increase in slag content over a longer period of time. This is because the external free water entering the interior of the matrix can alleviate the negative effect of phosphogypsum particles; therefore, the density of the matrix will increase with an increase in slag content, and the porosity will decrease accordingly (see Figure 9), which will increase the difficulty of the water exchange between the exterior and the interior of the system, reducing the mitigation effect of water on the negative effect of phosphogypsum particles. In conclusion, the mechanical stability of geopolymer first increases and then decreases with an increase in slag content. From Figure 13 and Figure 15, it can be seen that the appearance of mortar specimens R45S40 (S90), R45S30 (W90) and R25S40 (W90) is very complete, and no obvious cracks are found. However, mortar specimens R45S30 (S90), R25S50 (S90) and R25S50 (W90) show obvious cracks. Therefore, it can be determined that the standard curing strength of mortar specimen R45S30 is relatively poor at a longer age, whereas the standard curing strength and water-curing strength of mortar specimen R25S50 are relatively poor at a longer age, which is consistent with the analysis results in Figure 14.

### 3.4. Microscopic Analysis

#### 3.4.1. XRD

Figure 16 shows the XRD diffraction patterns of geopolymers with different phosphogypsum contents. From Figure 16, it can be seen that specimen P25R35 has a unique diffraction peak with an AFt peak compared to specimen P0R60. This shows that the new hydration product AFt is generated in an alkaline environment with the addition of phosphogypsum. Under proper alkaline conditions, the free phosphogypsum reacts with minerals containing calcium and aluminum to form AFt crystals, and the solid volume expands about 1.25 times, which has an important impact on the volume change of the specimen [9]. In addition, there are gypsum diffraction peaks in specimen P25R35 that are not present in specimen P0R60. The gypsum diffraction peak is the peak diffracted from phosphogypsum, which is the main component of phosphogypsum. This indicates that after 28 days of curing, there are still some phosphogypsum residues in specimen P25R35. The peak intensity of the CSH diffraction peak in P25R35 is slightly enhanced, indicating that phosphogypsum promoted the formation of CSH gel. There are diffraction peaks unique to regenerated fine powders, such as quartz and feldspar, in specimens P0R60 and P25R35, indicating that after 28 days of curing, there are residual recycled fine powders in specimens P0R60 and P25R35. Meanwhile, it can also be observed from Figure 16 that the diffraction peaks unique to recycled fine powders, such as quartz and feldspar, in specimen P25R35 show little change in peak intensity compared to specimen P0R60. This is because the solubility of crystalline quartz and calcite in an alkaline environment is relatively low, which has little impact on the diffraction intensity [47]. 

Compared to specimen P25R35, the gypsum diffraction peak, AFt diffraction peak, and C-S-H diffraction peak in specimen P60R0 significantly increased. This shows that phosphogypsum was in excess at this time, and most of them did not participate in the reaction. However, the increase in phosphogypsum content significantly increased the formation of hydration products, such as AFt and C-S-H. For the quartz diffraction peak, the peak intensity of specimen P60R0 significantly weakened, but most of this was caused by the dilution effect [54].

#### 3.4.2. FTIR

Figure 17 shows the infrared spectra of geopolymers with different phosphogypsum contents. As shown in Figure 17, the FTIR peak changes and shifts of geopolymers of different phosphogypsum contents mainly occur near 3400 cm^−1^ and between 400 and 2000 cm^−1^. Near the wave number of 3400 cm^−1^, the peak strength of specimen P60R0 is significantly stronger than that of specimen P25R35, whereas the peak strength of specimen P25R35 is significantly stronger than that of specimen P0R60. For the spectral band at 3400 cm^−1^, it corresponds to the –OH stretching vibration, indicating an increase in the hydroxyl groups. This shows that with an increase in phosphogypsum content, more Ca(OH)_2_ is generated from phosphogypsum recycled fine powder-based multi-source solid waste land polymer, which provides the conditions for generating more AFt and C-S-H gel. The spectral band near the wave number 1650 cm^−1^ is closely related to the bending vibration of H-O-H and bound water [55]. The peak strength of specimen P60R0 is the strongest here, followed by that of specimen P25R35, and the peak strength of specimen P0R60 is the weakest. This shows that with an increase in phosphogypsum content, more hydration products are generated from the phosphogypsum recycled fine powder-based multi-source solid waste geopolymer. This is because phosphogypsum has higher active ingredients (the part that participates in the hydration reaction), so more hydration products are generated. The spectral band at wave number 1460 cm^−1^ corresponds to the asymmetric stretching vibration of the C–O bond, which is related to the carbonization of the cementitious material [56]. The peak intensities of samples P0R60, P25R35 and P60R0 near this wave number range increase in turn. Compared to phosphogypsum, the recycled fine powder consumes more NaOH. With an increase in phosphogypsum content, the NaOH consumption of the slurry decreases, the residual NaOH increases, and the CO_2_ consumed in the carbonization reaction increases. The spectral band at wave number 1115 cm^−1^ corresponds to the anti-symmetric stretching vibration of 0-S-O, which is correlated with [57]. The peak strength of P60R0 is significantly stronger than that of P25R35, indicating that a large amount of phosphogypsum remains in P60R0, which may lead to the formation of a large amount of delayed AFt in the later period, affecting the later strength of the slurry. The spectral band near 1000 cm^−1^ corresponds to the stretching vibration of Si-O-Si and the asymmetric stretching vibration of Si-O-T (where T is Si or Al) [46,58]. The peak intensity of sample P25R35 here is slightly weaker than that of sample P0R60, but this does not mean that phosphogypsum promotes the dissolution of recycled fine powder because it may also be caused by dilution due to the reduction in recycled fine powder content. However, compared to specimen P0R60, the peak strength of specimen P25R35 moves towards the direction of low wave number, which indicates the dissolution of aluminate and the formation of low polymerization degree hydration gels, such as NASH and CASH [59,60]. Therefore, phosphogypsum can promote the dissolution of regenerated fine powder to participate in the reaction to generate geopolymeric compounds. It should be noted that although phosphogypsum promotes the dissolution of recycled fine powder to participate in the reaction, the final gelation of NASH, CASH, etc. will also be reduced because of the reduction in the total amount of recycled fine powder.

#### 3.4.3. SEM

Figure 18 shows the SEM diagram of geopolymers with different phosphogypsum contents. From Figure 18a–c, it can be clearly observed that specimen P0S40—M1.3N6 has smaller pores and the densest structure, followed by specimen P25R35S40-M1.3N6 and specimen P60S40—M1.3N6, which has the loosest structure. According to the analysis of XRD and FTIR, an increase in phosphogypsum content can promote the geopolymer to generate more hydration products. However, for NASH, CASH and other low degrees of polymerization hydration gels, the content of specimen P0R60 is the highest, and the content of specimen P25R35S40 M1.3N6 is small but exists, whereas the content of specimen P60S40—M1.3N6 is absent. This shows that hydration products, such as NASH and CASH, are the main factors affecting the structural compactness of phosphogypsum and recycled fine powder-based multi-source solid waste geopolymers. From Figure 18d–f, it can be seen that compared to specimen P0S40—M1.3N6, the hydration products and some partially unreacted raw material particles in specimen P25R35S40 M1.3N6 and P60S40—M1.3N6 are not tightly overlapped with each other, especially in specimen P60S40—M1.3N6. Due to the lack of tight internal overlap, the specimen structure becomes loose, which has a significant impact on the structural density and strength. At the same time, it can be seen from Figure 18e,f that the gypsum particles do not fully react, in which the phosphogypsum particles in specimen P60S40—M1.3N6 are more than the phosphogypsum particles in specimen P25R35S40 M1.3N6, further supporting the analysis of XRD and FTIR. In addition, hydration products such as NASH and CASH are generated in specimen P0S40 M1.3N6 (see Figure 18g), and AFt is generated in specimens P25R35S40 M1.3N6 and P60S40 M1.3N6 (see Figure 18b,h,i), which can be verified in Figure 18. At the same time, it can be seen that the AFt in specimen P25R35S40 M1.3N6 is mainly short columnar, whereas the AFt in specimen P60S40 M1.3N6 is slender, indicating that recycled fine powder can promote the transformation of AFt from slender to short columnar, thereby improving the density and strength of the slurry.

### 3.5. Stable and Synergistic Mechanism of Multi-Source Solid Waste

The main chemical component of phosphogypsum is ${\mathrm{CaSO}}_{4}\cdot 2H_{2}O$, which easily dissolves a large amount of ${\mathrm{SO}}_{4}^{2-}$ and Ca^2+^ in an alkaline environment. It is the main source ${\mathrm{of}\;\mathrm{SO}}_{4}^{2-}$ and one of the sources of Ca^2+^. The composition of the regenerated fine powder is relatively mixed, but its silicon aluminum oxide proportion exceeds half, making it an appropriate silicon aluminum material. Under the excitation of an alkaline activator, active [AlO_4_]^5−^ and [SiO_4_]^4−^ can be released. Slag contains a large amount of active CaO, which is the main source of Ca^2+^ in the system. The ${\mathrm{SO}}_{4}^{2-}$, [AlO_4_]^5−^, [SiO_4_]^4−^ and Ca^2+^ participate in a series of hydration reactions, generating CSH, CASH, NASH and other hydration gels and AFt crystals [61,62,63,64]. These hydration products directly or indirectly determine the stability of the geopolymer. The hydration reaction abstract diagram inside the system is shown in Figure 19.

The main hydration reactions involved in the phosphogypsum and recycled fine powder-based multi-source solid waste geopolymer can be abbreviated as Formulas (7)–(10).
(7)$${\mathrm{Ca}}^{2+}+\mathrm{Si}-O+{\mathrm{OH}}^{-}+H_{2}O\to C-S-H+H_{2}O$$
(8)$${\mathrm{Ca}}^{2+}+{\left[{\mathrm{AlO}}_{4}\right]}^{4-}+{\left[{\mathrm{SiO}}_{4}\right]}^{4-}+{\mathrm{OH}}^{-}+H_{2}O\to C-A-S-H$$
(9)$$C-A-S-H+{\mathrm{OH}}^{-}+{\mathrm{Na}}^{+}\to N-A-S-H$$
(10)$${\mathrm{Ca}}^{2+}+{\mathrm{Al}}^{3+}+{\mathrm{OH}}^{-}+{\mathrm{SO}}_{4}^{2-}+H_{2}O\to 3\mathrm{CaO}\cdot {\mathrm{Al}}_{2}O_{3}\cdot 3{\mathrm{CaSO}}_{4}\cdot 32H_{2}O\;(\mathrm{AFt})$$

According to the analysis in Section 3.1, the compressive strength *σ,* porosity *P*, pore saturation *S* and AFt generation determine the volume stability of geopolymers. According to the analysis in Section 3.4.1, the raw materials in the geopolymer will not react completely, and there is a large number of inert or non-participating particle residues in the system, such as recycled fine powder particles and phosphogypsum particles. These unreacted solid particles overlap with each other, forming the basic framework of the geopolymer. Raw materials are the sources of various ions within the system, directly controlling the ion concentration inside the system, thereby affecting the hydration process of Equations (7)–(10) and promoting or inhibiting the generation of hydration gel products, such as CSH, CASH, NASH and AFt. These hydration products are the source of slurry strength and directly determine the compressive strength of the slurry. At the same time, these hydration products can also fill the pores of the slurry, thereby affecting its compactness, porosity *P* and pore saturation *S*. When the slag content is fixed, with the increase in phosphogypsum content, the amount of recycled fine powder will be reduced accordingly. On the one hand, with the increase in phosphogypsum content, the concentration of ${\mathrm{SO}}_{4}^{2-}$ and Ca^2+^ in the system promotes hydration reactions (7) and (10) to the left, and more CSH gel and AFt crystals are generated (Figure 16 can prove this). On the other hand, as the amount of recycled fine powder decreases, the concentrations of [AlO_4_]^5−^ and [SiO_4_]^4−^ inside the system will correspondingly decrease, further inhibiting the hydration and gelation generation of CASH, NASH and other materials. When the amount of phosphogypsum is fixed, the amount of slag will increase, and the amount of recycled fine powder will decrease. The Ca^2+^ inside the slurry significantly increases, whereas the concentrations of [AlO_4_]^5−^ and [SiO_4_]^4−^ significantly decrease. The hydration reactions of Equations (7) and (10) within the system will proceed to the left, whereas the reactions of Equations (8) and (9) are suppressed. The final manifestation is an increase in the generation of CSH and AFt and a decrease in the generation of CASH and NASH. It should be noted that the active component of slag is much higher than that of phosphogypsum and recycled fine powder, so an increase in slag will lead to a significant increase in CSH cementation. In addition, ${\mathrm{SO}}_{4}^{2-}$ is the decisive factor that determines the amount of AFt generation, so an increase in Ca^2+^ will lead to more AFt generation, but it will not lead to a sharp increase in AFt generation [46]. According to the analysis in Section 3.2, it can be concluded that the contact between ${\mathrm{CaSO}}_{4}\cdot 2H_{2}O$ and water in the external environment is a key factor affecting the water stability of geopolymers. When the slag is fixed, the recycled fine powder will increase and phosphogypsum will decrease accordingly. On the one hand, promoting hydration reactions of Equations (8) and (9) generates more hydration gel products, such as CASH and NASH, improves the density of the matrix and reduces the possibility of water entering the system from the external environment. On the other hand, ${\mathrm{CaSO}}_{4}\cdot 2H_{2}O$ content in the system will decrease with a decrease in phosphogypsum content, and the slag promotes the rightward progression of Equations (7) and (10), generating more CSH cementitious material and AFt. On the other hand, the reduction in recycled fine powder inhibits the progression of Equations (8) and (9), resulting in a decrease in the generated hydration, such as CASH and NASH. However, the active component of slag is higher than that of phosphogypsum and recycled fine powder, so the increased CSH cementation can completely offset the negative impact of CASH and NASH reduction on the matrix structure, significantly improving the density of the matrix and blocking the entry of external water.

According to the analysis in Section 3.3, it can be concluded that the expansion stress caused by the delayed AFt and the constraint stress generated by the development of its strength are the main factors affecting the long-term stability of geopolymers. When the slag is fixed, the recycled fine powder will increase and phosphogypsum will decrease accordingly. On the one hand, recycled fine powders promote hydration reactions and generate more hydration binders, such as CASH and NASH, increasing the constraint stress generated by strength development. On the other hand, the amount of delayed AFt decreases with a decrease in phosphogypsum content. When phosphogypsum is fixed, the slag will increase and the recycled fine powder will decrease. The change in the amount of delayed AFt generation within the matrix is not significant, but slag can significantly affect the constraint stress generated by its strength development.

## 4. Conclusions

The stability and synergistic mechanism under the combined action of multi-source solid waste, such as phosphogypsum, recycled fine powder and slag, were studied. Based on these results and analyses, the following conclusions are obtained:(1)The combined action of phosphogypsum, recycled fine powder, slag and other multi-source solid wastes can not only control the generation of AFt crystals but also control the capillary stress inside the matrix by adjusting the pore structure of the matrix, thus improving the volume stability of phosphogypsum and recycled fine powder-based multi-source solid waste geopolymers, reducing its volume deformation, and avoiding the expansion and damage of geopolymers. The volume of specimen P35R25 slightly shrinks without any risk of expansion or damage. Meanwhile, the 154-day volume change rate of specimen P35R25 decreased by 66.4% compared to that of specimen P0R60, and its 28-day volume change rate was 10.7% lower than that of specimen OPC.(2)The synergistic effect of phosphogypsum, recycled fine powder and slag promotes the generation of more CASH and NASH hydration gels, improves the pore structure of the matrix and reduces the negative impact on the matrix, thus improving the water stability of phosphogypsum and recycled fine powder-based multi-source solid waste geopolymer. The water stability of mortar specimen P15R45 is the best, with a softening coefficient of 1.06, which is 26.2% higher than that of mortar specimen P35R25.(3)The synergistic effect of phosphogypsum, recycled fine powder and slag reduces the negative impact of delayed AFt and improves the mechanical stability of the geopolymer. The 118-day standard curing compressive strength (28 days of sealing curing plus 90 days of standard curing) of mortar specimen P15R45 decreased by only 2.3% compared to its 28-day compressive strength, whereas its 118-day water-curing compressive strength (28 days of sealing curing plus 90 days of immersion curing) increased by 18.2% compared to its 28-day compressive strength.(4)Microscopic analysis shows that phosphogypsum can promote the generation of more hydration products, especially AFt, which has an important impact on the pore structure and volume change of the slurry. Both phosphogypsum and recycled fine powder in geopolymers do not completely react, and the unreacted phosphogypsum particles and recycled fine powder particles not only have an important impact on the compactness of the matrix structure but also affect its stability. Recycled fine powder can promote the generation of hydration products, such as CASH and NASH, from geopolymers, improve the compactness of the slurry and enhance its stability. When the content of phosphogypsum increases, the content of recycled fine powder decreases, and the compact structure of the geopolymer slurry is destroyed, thereby becoming loose and porous.

To sum up, geopolymers have excellent stability due to the combined action of multi-source solid waste, such as phosphogypsum, recycled fine powder and slag. The contents shown in this study can be used as an effective method to change hazardous materials into green building materials and reduce harm to the environment.

## Figures and Tables

**Figure 1 polymers-15-02696-f001:**
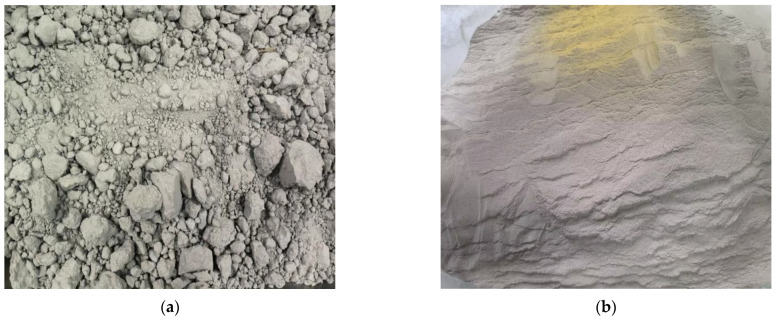
Appearance and morphology of phosphogypsum. (**a**) Original phosphogypsum; (**b**) phosphogypsum powder.

**Figure 2 polymers-15-02696-f002:**
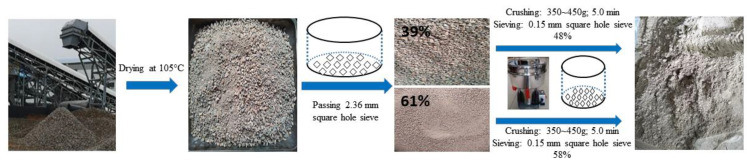
Flow chart of recycled construction waste fine powder preparation.

**Figure 3 polymers-15-02696-f003:**
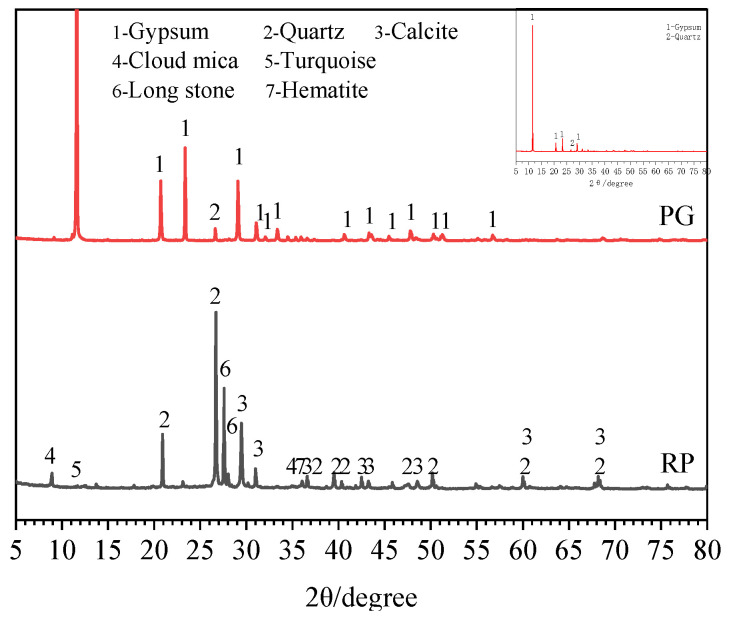
XRD of the phosphogypsum and recycled construction waste fine powder.

**Figure 4 polymers-15-02696-f004:**
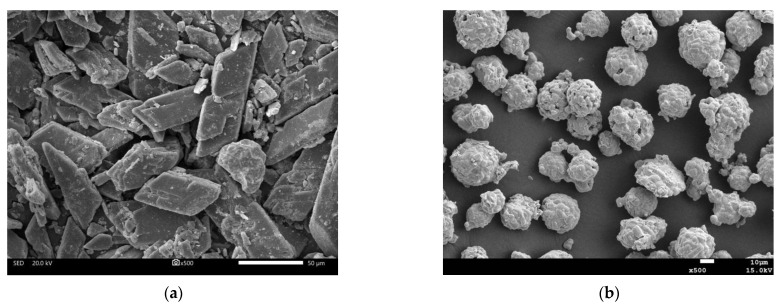
SEM of the phosphogypsum and recycled construction waste fine powder. (**a**) PG; (**b**) RP.

**Figure 5 polymers-15-02696-f005:**
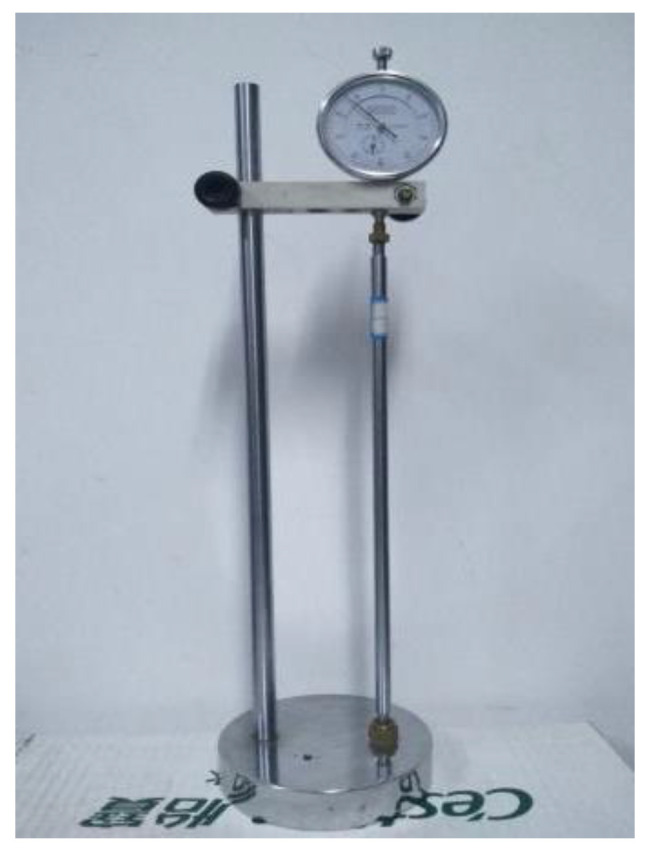
BC-300 cement length meter.

**Figure 6 polymers-15-02696-f006:**
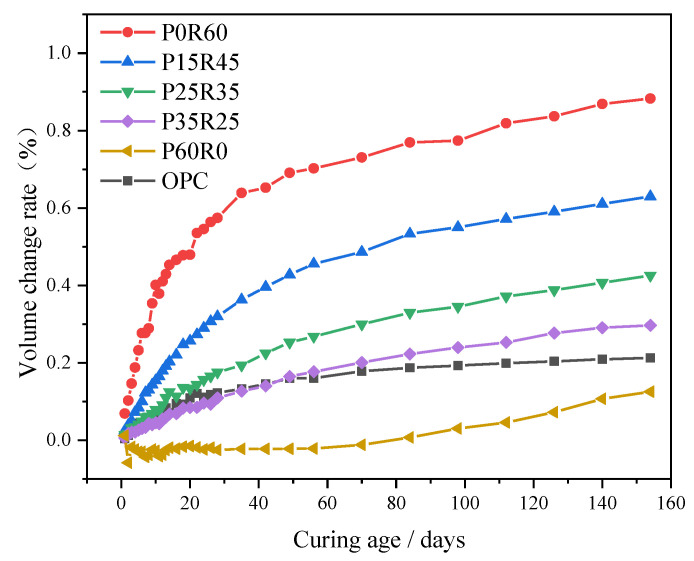
Effect of phosphogypsum content on the volume change rate of geopolymers.

**Figure 7 polymers-15-02696-f007:**
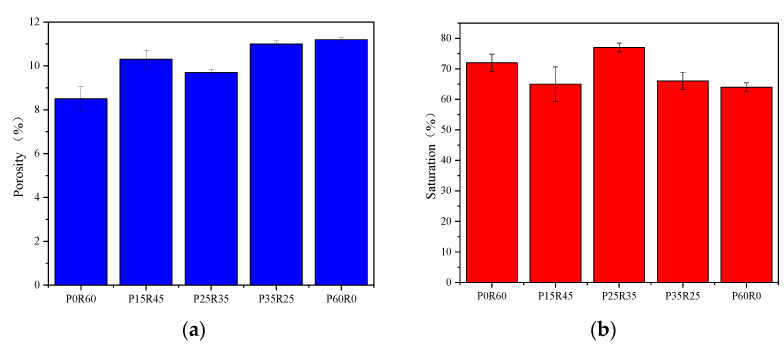
Effect of phosphogypsum content on porosity and pore saturation of geopolymers. (**a**) Porosity; (**b**) pore saturation.

**Figure 8 polymers-15-02696-f008:**
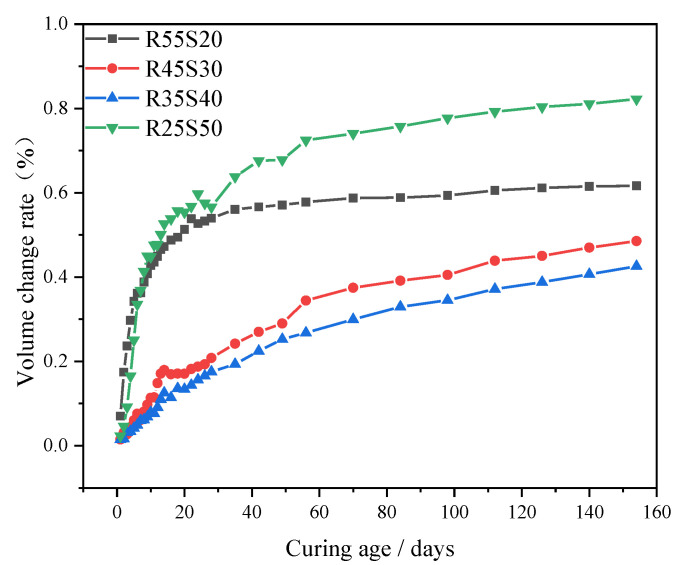
Effect of slag content on the change rate of geopolymer volume.

**Figure 9 polymers-15-02696-f009:**
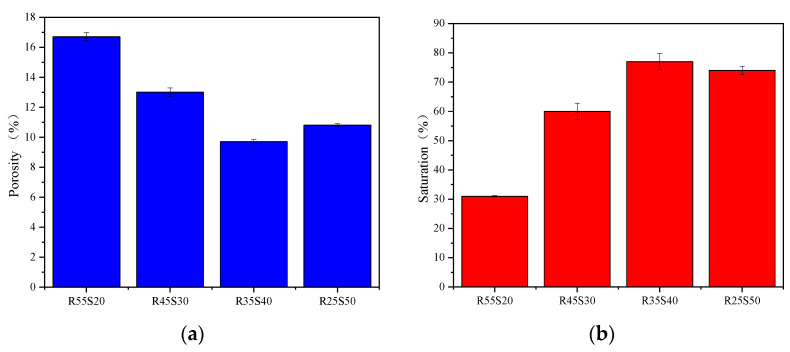
Effect of phosphogypsum content on geopolymer porosity and pore saturation. (**a**) Porosity; (**b**) pore saturation.

**Figure 10 polymers-15-02696-f010:**
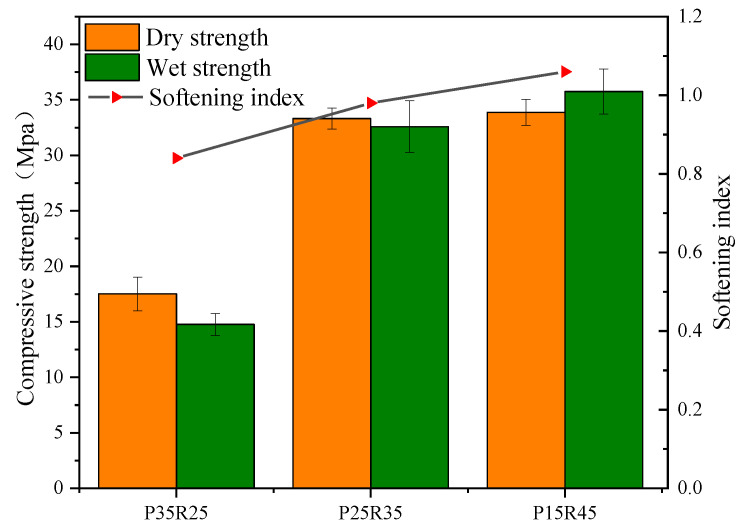
Effect of recycled fine powder content on water stability.

**Figure 11 polymers-15-02696-f011:**
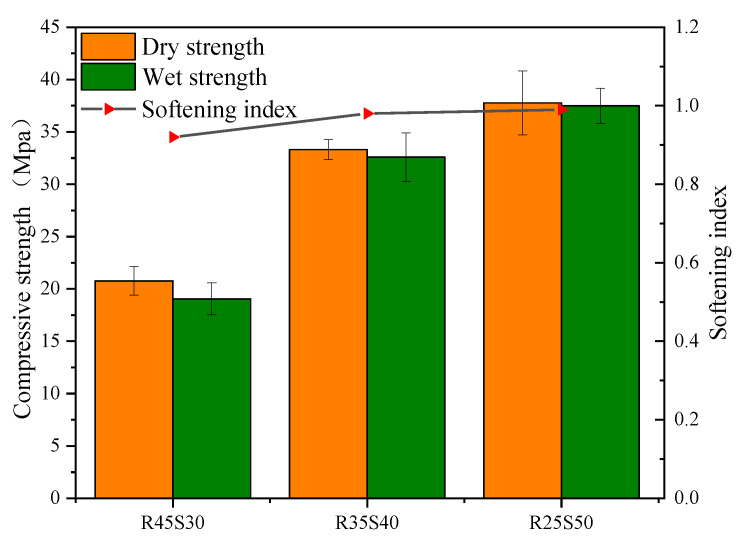
Effect of slag content on water stability.

**Figure 12 polymers-15-02696-f012:**
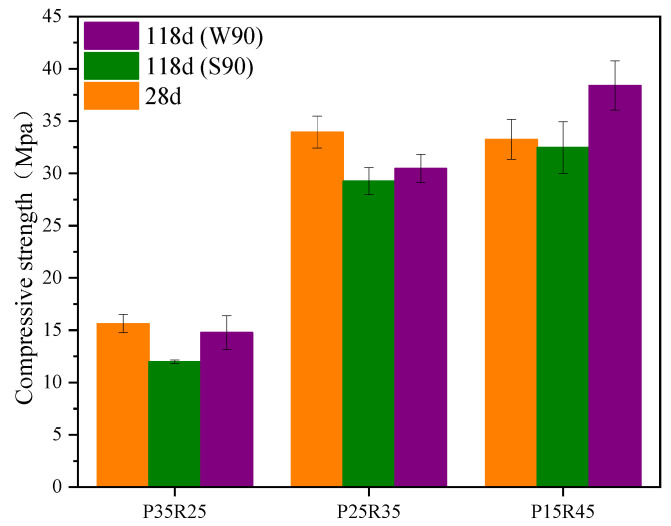
Effect of recycled fine powder on mechanical stability.

**Figure 13 polymers-15-02696-f013:**
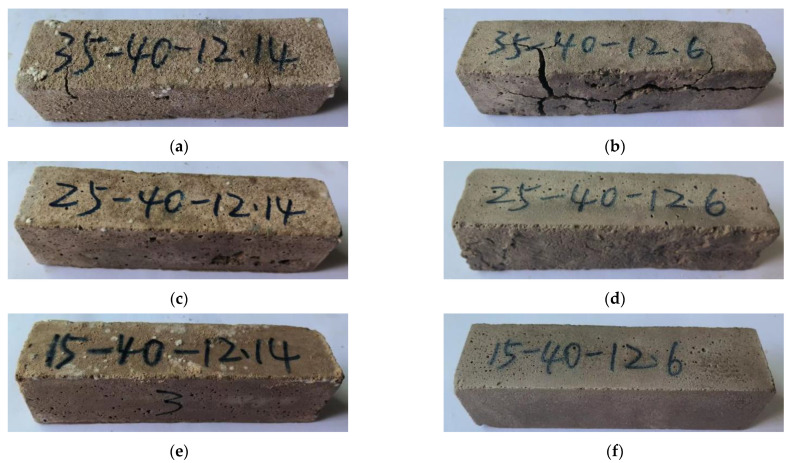
Appearance morphology of mortar specimens with different recycled fine powder contents after standard curing/water curing for 90 days. (**a**) P35R25 (S90); (**b**) P35R25 (W90); (**c**) P25R35 (S90); (**d**) P25R35 (W90); (**e**) P15R45(S90); (**f**) P15R45(W90).

**Figure 14 polymers-15-02696-f014:**
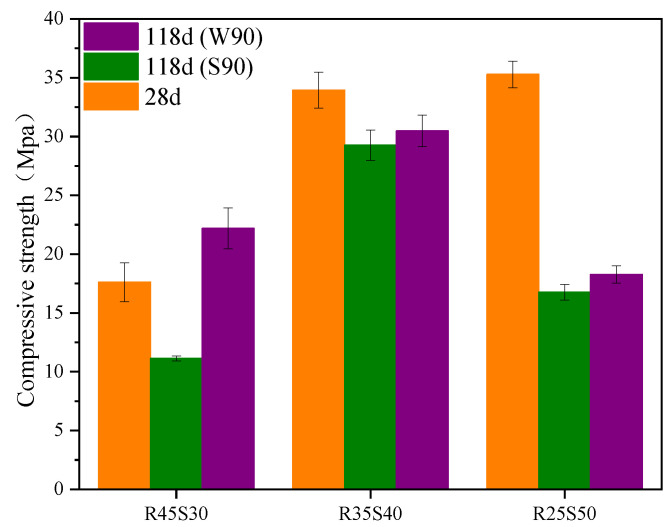
Effect of the slag content on mechanical stability.

**Figure 15 polymers-15-02696-f015:**
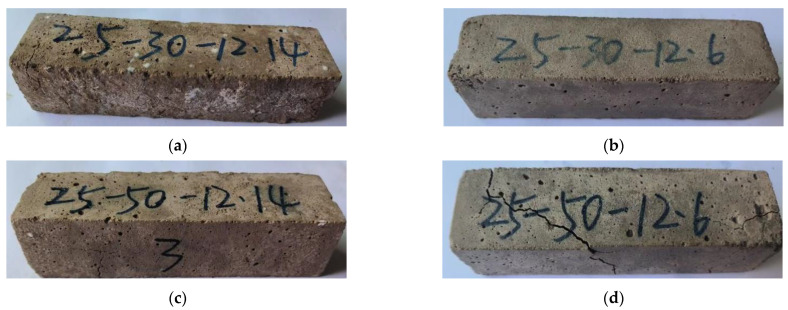
Appearance morphology of mortar specimens with different slag contents after standard curing/water curing for 90 days. (**a**) R45S30 (S90); (**b**) R45S30 (W90); (**c**) R25S50 (S90); (**d**) R25S50 (W90).

**Figure 16 polymers-15-02696-f016:**
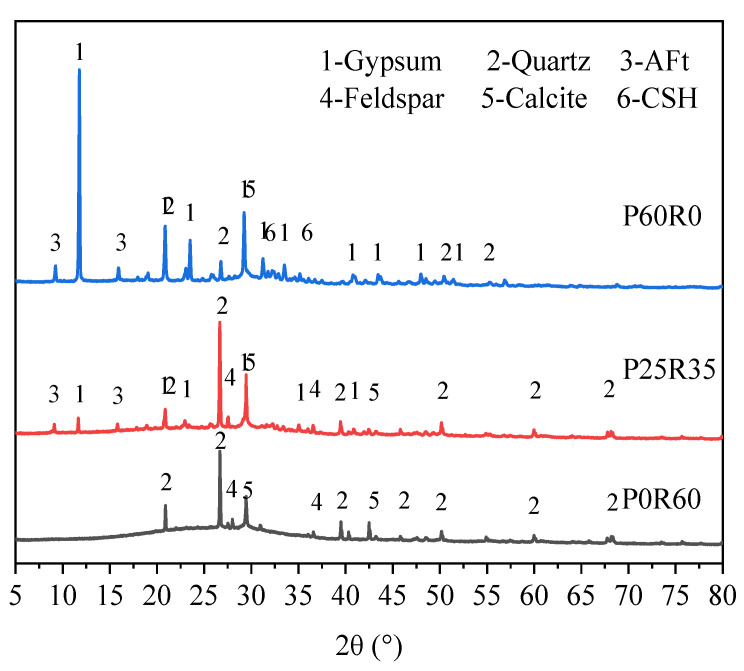
28-day XRD patterns of geopolymers with different phosphogypsum contents.

**Figure 17 polymers-15-02696-f017:**
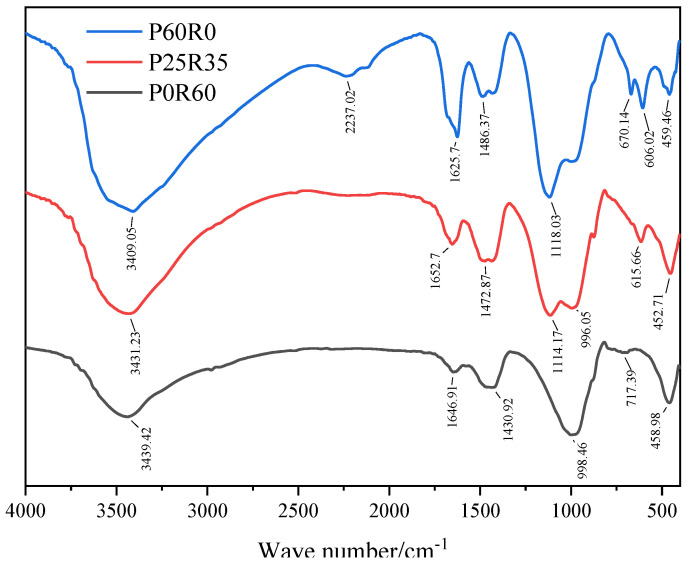
28-day infrared spectra of geopolymers with different phosphogypsum contents.

**Figure 18 polymers-15-02696-f018:**
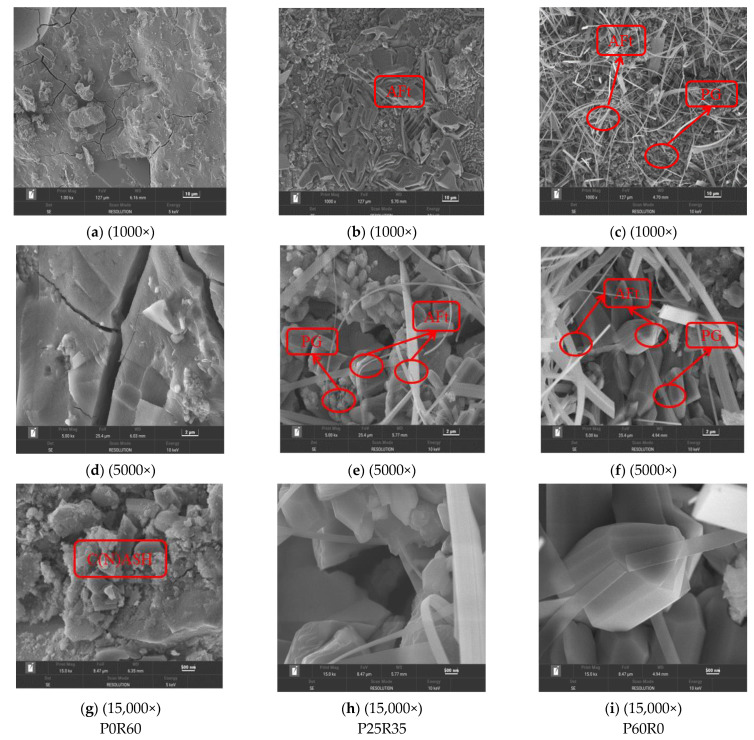
28-day SEM diagram of geopolymers with different phosphogypsum contents.

**Figure 19 polymers-15-02696-f019:**
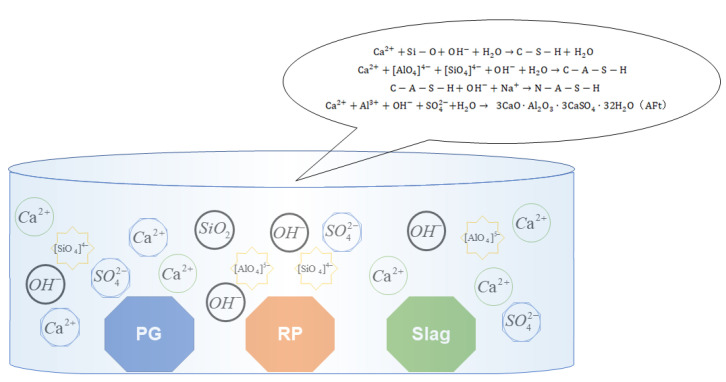
Abstract diagram of hydration reaction inside phosphogypsum and recycled fine powder-based multi-source solid waste geopolymer.

**Table 1 polymers-15-02696-t001:** Chemical composition of the major raw materials %.

	$SiO_{2}$	$Al_{2}O_{3}$	CaO	$SO_{3}$	F	$Fe_{2}O_{3}$	$P_{2}O_{5}$	MgO	$Na_{2}O$	$TiO_{2}$	MnO	$K_{2}O$	Others
PG	4.3	0.4	33.8	41.1	1.8	0.4	1.0	0.1	0.1	0.1	0.1	0.1	16.7
RP	49.4	20.2	17.3	0.9	/	4.7	0.1	1.4	1.3	0.6	0.1	2.1	1.9
Slag	31.2	8.4	38.6	0.2	/	7.6	0.1	3.3	0.3	0.2	2.6	0.3	7.2

**Table 2 polymers-15-02696-t002:** Main technical indicators of cement.

Specific Area (m^2^/kg)	Setting Time (min)		Flexural Strength (Mpa)		Compressive Strength (Mpa)	
	Initial	Final	3 d	28 d	3 d	28 d
363	161	234	5.6	8.4	25.4	49.8

**Table 3 polymers-15-02696-t003:** Particle grading of fine river sand.

Sieve Size (mm)	Residue (g)	Percentage of Residue (%)	Accumulated Residue Percentage (%)	Fineness Modulus
2.36	216.82	18.8	18.8	2.96
1.18	157.94	13.7	32.5	
0.6	331.64	28.7	61.2	
0.3	317.04	27.5	88.6	
0.15	73.88	6.4	95.0	

**Table 4 polymers-15-02696-t004:** Industrial sodium silicate solution parameters.

Model	SiO_2_ Content (wt.%)	Na_2_O Content (wt.%)	Density (Be °C)	Modulus (M)
BE40-6	8.5	26.5	40	3.2

**Table 5 polymers-15-02696-t005:** Mix proportion of phosphogypsum recycled fine powder-based multi-source solid waste geopolymer slurry.

Mix ID	Powder Material (wt.%)				Alkali Activator		W ^c^/B ^d^
	PG	RP	Slag	OPC	Ms ^a^	N ^b^ (%)	
OPC				100			0.42
P0R60		60	40		1.3	6	0.42
P15R45	15	45	40		1.3	6	0.42
P25R35 (R35S40)	25	35	40		1.3	6	0.42
P35R25	35	25	40		1.3	6	0.42
P60R0	60		40		1.3	6	0.42
R55S20	25	55	20		1.3	6	0.42
R45S30	25	45	30		1.3	6	0.42
R25S50	25	25	50		1.3	6	0.42

^a^ Ms = SiO_2_/Na_2_O modulus; ^b^ N = Na_2_O/Powder material; Na_2_O = Na_2_O (within SS) + Na_2_O (obtained from SH); 2NaOH (1.29 g) → Na_2_O (1 g) + H_2_O (0.29 g); ^c^ B (Binder) = PG + RMP + Slag + SH + solid part of SS; ^d^ W (Water content) = water part of SS + additional water.

**Table 6 polymers-15-02696-t006:** Mix proportion of phosphogypsum recycled fine powder-based multi-source solid waste geopolymer mortar.

Mix ID	Powder Material (wt.%)			^a^ W-R	Alkali Activator		W/B	B/^b^ S
	PG	RP	Slag	wt.%	Ms	N (%)		
P35R25	35	25	40	1	1.3	6	0.48	1:2
P25R35 (R35S40)	25	35	40	1	1.3	6	0.48	1:2
P15R45	15	45	40	1	1.3	6	0.48	1:2
R45S30	25	45	30	1	1.3	6	0.48	1:2
R25S50	25	25	50	1	1.3	6	0.48	1:2

^a^ W-R = Quality of water reducer; ^b^ S = Quality of sand.

## Data Availability

Data will be made available on request.

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
