# Peer review of "The Synergistic Mechanism and Stability Evaluation of Phosphogypsum and Recycled Fine Powder-Based Multi-Source Solid Waste Geopolymer"

_polymers, 2023, doi:10.3390/polym15122696_

Round 1
Reviewer 1 Report
Dear Authors!
This paper examines the effect of the synergistic mechanism of solid wastes such as phosphogypsum, recycled fine powder and slag on the properties of a geopolymer material. The analyzed problem is of scientific and technical interest for the field of materials, engineering and environmental sustainability. The manuscript may be published in this journal if the following comments are fully taken into account in the revised version.
Сomments
1. Incorrectly formatted links in the test: the first link [1] is replaced by [50], some are superscripted, for example, links 36 - 41. Also, the references do not correspond to reality, for example, in references 17 - 19 in the article, some authors are indicated, and others in the reference list (Zhang Xiuqin et al. [17]… .References: [17] Król, M., Rożek, P., Chlebda, D., Mozgawa, W., 2018. Influence of alkali metal cations/type of activator on the structure of alkali-activated fly ash - ATR-FTIR studies.) All citations should be checked.
2. “The main component of MSW phosphogypsum is CaSO4∙2H2O”, “The main chemical component of phosphogypsum is CaSO4∙2H2O”. Perhaps, in the second case, they met with the proposal that appeared and reformulated ".
3. From the sentence “When the geopolymer with phosphogypsum is prepared, delayed ettringite (AFt) can be continuously generated, which may damage the basic structure of the geopolymer, leading to expansion damage of the material”, it is not clear how ettringite can damage the basic structure of the geopolymer? It is necessary to refer either to a literary source, or to provide explanations.
4. The paper says, “Phosphogypsum was dried at 40℃ to constant weight, crushed manually, and passed through a 0.15mm square mesh sieve to obtain phosphogypsum powder with particle size less than 0.16mm.” Passing particles through a sieve with a mesh size of 0.15 mm, the authors of the article obtain particle sizes less than 0.16 mm, but the wording “particle sizes of 0.15 mm or less” would be more correct.
5. Table 6 does not match the designation of the samples in accordance with the content of each material. For example, for sample number P35R25, it is obvious that P is the content of phosphogypsum and it should be 35 wt.%, and not 15% as indicated in the table.
6. On page 12, the sentence "It can be thought that the slag promotes the hydration reaction in the system, generating more hydration hydrated such as CASH, CASH, NASH and Aft ..." mentions CASH twice, probably CASH is confused with CSH.
7. Page 13 says "Meanwhile, as shown in Figure 9, the maximum difference in slurry porosity is 6.0%...", but the reported difference in slurry porosity is 7.0%.
8. On page 14, the sample is called "P35P25S40M1" before it was P35P25. It is necessary to bring the designations to a single style.
9. On the radiograph (Figure 16), it is necessary to indicate the peaks found in the P0R60 sample, as well as in the P60R0 and P25R35 samples.
Reviewer 2 Report
Please see the comments below:
Abstract:
-lines 10-11: what is phosphogypsum? Is this recycled material? Also, what do you mean by recycled fine powder? Is it fly ash or slag or something else? Do all recycled powders use to produce geopolymer or only those with high silica and alumina contents? at which conditions the phosphogypsum-based geopolymer might undergo a risk of expansion cracking? – same question for recycled fine powder-based geopolymer? What application this paper suggested?
- What are the main tests used in the paper?
- does the author test geopolymer in the format of mortar? Was it mixed with sand? What are the test conditions?
Lines 19-22: define CaSO4∙2H2O, AFt
Introduction:
- Restructure for this section is required. Line 39 mentioned geopolymer without a definition. What is geopolymer? Advantages of geopolymer? ingredients? Issues with current mixtures? Problems with phosphogypsum-based geopolymer…etc.
- Lines 67-68: “The shrinkage problem of geopolymers is one of the main reasons affecting their practical engineering applications [20,21]”. Which types of geopolymer and under which conditions? It is well known in the literature that geopolymers based on fly ash and slag are highly stable as evidenced by many research (example: Al Bakri, Mohd Mustafa, et al. "Review on fly ash-based geopolymer concrete without Portland Cement." Journal of Engineering and technology research 3.1 (2011): 1-4.)
Raw materials and experimental methods
The ingredients of geopolymer are not clear. Different materials were listed in this section (Phosphogypsum, Recycled construction waste fine powder, Slag, Ordinary Portland cement OPC, Sodium hydroxide and Sodium silicate solution) without mentioning the role of each material in the geopolymer mixture. Mix proportions are not clear.
Lins 170-172: “ accurately weighed 170 various powder materials according to the mix proportion in the experiment, mixed and 171 stirred them evenly” – Did the author mix all mentioned powder materials in the geopolymer mix for this research? Does this include OPC (cement) mentioned in line 152?
Extensive proofreading is required - the writing style must be improved.
Round 2
Reviewer 2 Report
The review recognised the author's responses. following are remain comments:
- Please mention in the abstract the main tests conducted (the reviewer realise that all tests were mentioned in section 2.2 of the paper - it is necessary to highlight conducted tests briefly in the abstract before presenting important findings.
- The current format of methodology is still confusing as tests were mentioned (section 2.2) before mixtures (section 2.3 - Tables 5 and 6). Please restructure the methodology section to include the following headings: materials used, geopolymer mixture/proportions, control mixtures, specimen preparations, curing conditions, and then testing.
Minor corrections are required.
Author Response
Manuscript ID: polymers-2434631
Title: The Synergistic Mechanism and Stability Evaluation of Phosphogypsum and Recycled Fine Powder Based Multi-Source Solid Waste Geopolymer
Journal: Polymers
Dear Reviewer,
Thank you very much for your valuable and helpful comments. Your suggestions are really valuable and helpful for revising and improving our paper. In order to make it easier for reviewers and editors to read, the authors did not use the "track mark" mode during the reference revision process. Besides that, all changes have been marked using the "track mark" function. According to your suggestions, we have revisied this manuscript. you can find it in attachment.
